# Device-Assessed Physical Activity and Sedentary Behaviors in Canadians with Chronic Disease(s): Findings from the Canadian Health Measures Survey

**DOI:** 10.3390/sports7050113

**Published:** 2019-05-16

**Authors:** Gabriel Hains-Monfette, Sarah Atoui, Kelsey Needham Dancause, Paquito Bernard

**Affiliations:** 1Department of Physical Activity Sciences, Université du Québec à Montréal, Montréal, QC H3C3P8, Canada; gabrielhainsmonfette@gmail.com (G.H.-M.); atoui.sarah@courrier.uqam.ca (S.A.); needham-dancause.kelsey@uqam.ca (K.N.D.); 2Research Center, University Institute of Mental Health at Montréal, Montréal, QC H1N3M5, Canada

**Keywords:** Chronic disease, physical activity, sedentary behavior, multimorbidity, health promotion

## Abstract

Physical activity and sedentary behaviors (SB) are major determinants of quality of life in adults with one or more chronic disease(s). The aim of this study is to compare objectively measured physical activity and SB in a representative sample of Canadian adults with and without chronic disease(s). The Canadian Health Measures Survey (CHMS) (2007–2013) was used in this study. Daily time spent in physical activities and sedentary behaviors were assessed by an accelerometer in Canadians aged 35–79 years. Data are characterized as daily mean time spent in moderate-to-vigorous physical activity (MVPA), light physical activity (LPA), steps accumulated per day and SB. Chronic diseases (chronic obstructive pulmonary disease, diabetes, heart diseases, cancer) were assessed via self-report diagnostic or laboratory data. Weighted multivariable analyses of covariance comparing physical activity and SB variables among adults without and with chronic disease(s) were conducted; 6270 participants were included. Analyses indicated that 23.9%, 4.9% and 0.5% had one, two, and three or more chronic diseases. Adults with two and more chronic diseases had significantly lower daily duration of MVPA and LPA, daily step counts, and higher daily duration of SB compared to adults without chronic diseases. Interventions targeting physical activity improvement and SB reduction might be beneficial for Canadian multimorbid adults.

## 1. Introduction

Chronic Diseases (CDs) are a leading cause of premature mortality in Canada. The most prevalent CDs in 2012 among Canadian adults included respiratory (9.4%) and cardiovascular diseases (8.1%), diabetes (7.6%), and cancer (2.5%) [1]. Average annual prevalence increased by 2.5%, 1.0%, 4.2%, and 1.1%, respectively [1], from 2000–2012.

Epidemiological studies generally analyze CDs individually. However, in 2012, 3.6% of Canadian adults [1] were multimorbid (i.e., diagnostic of two or more chronic diseases) [2]. Prevalence increases with age; in 2012, 0.9% of Canadians aged 35–49 were multimorbid, 3.7% of those aged 50–64, 11.0% of those aged 65–79, and 13.9% of those aged 80 and over [1]. Rates of multimorbidity may thus dramatically increase with the aging of Canada’s population over the next 20 years. The development of interventions to prevent multimorbidity is thus urgently needed [1]. In this context, a first international report of the UK’s Academy of Medical Sciences recommended the identification of factors associated with multimorbidity as a research priority [2].

Physical inactivity is defined as “an insufficient physical activity level to meet present physical activity recommendations” [3]. Sedentary behavior is defined as “any waking behavior characterized by an energy expenditure ≤1.5 metabolic equivalents while in a sitting, reclining or lying posture” [3]. These are two health behaviors known to contribute to a number of CDs. Higher levels of physical activity at low, moderate, or vigorous intensity were associated with higher levels of health-related quality of life and better symptom management in adults with chronic obstructive pulmonary disease (COPD), cardiovascular diseases, diabetes, and cancer [4]. Sedentary behaviors were related to deleterious health outcomes and higher risk of hospitalization in adults with a single CD [5]. The negative association between time spent in sedentary behaviors and health are relatively independent of physical activity [6]. Consequently, physical activity and sedentary represent distinct behaviors to target for the management of symptoms and health outcomes in adults with CDs [7,8,9]. 

Although physical activity and sedentary behaviors have been extensively studied among adults with single CDs, few studies have assessed these behaviors in multimorbid adults. An inverse cross-sectional association between self-reported physical activity and multimorbidity has been found in Atlantic Canadian [10], European [11,12], and international surveys [13]. The magnitude of the association between physical activity and number of CDs was stronger for men [10]. Self-reported physical inactivity was also significantly associated with increased odds of multimorbidity in middle-aged women over 20 years of follow-up [14]. However, physical activity measured with questionnaires is generally overestimated in comparison to accelerometer measures, particularly in adults with CD [15,16,17]. Recently, two analyses of National Health and Nutrition Examination Survey (NHANES) data suggested that higher daily or weekly minutes of accelerometer-measured moderate to vigorous physical activity (MVPA) was related to lower multimorbidity risk [18,19]. A negative linear association between self-reported sedentary behaviors (e.g., television viewing) and number of CDs has been found in two international surveys [11,12,20].

To date, a comparison of objectively measured physical activity data between adults without CD or with one or more CDs has not been examined in a Canadian representative population-based study. Time spent in objectively measured sedentary behaviors has not been investigated in adults with multiple CDs. Moreover, previous investigations characterized multimorbidity by adding a CD diagnostic with a recurrent medical condition (e.g., diabetes with obesity and elevated total cholesterol) [12]. However, Ording et al. [21] defined multimorbidity as the existence of two or more diagnostics of chronic disease. Our objective in this study was to compare device-measured physical activity and sedentary behaviors of Canadian adults with one CD or multimorbidity to those without CDs in a nationally representative sample. 

## 2. Materials and Methods

Data from cycles 1, 2, and 3 (2007–2013) of the Canadian Health Measures Survey (CHMS) were used for this study. The CHMS is a multi-wave, national survey including people from 3 to 79 years of age across 10 provinces and using a stratified three-stage sampling strategy [22]. Data collection was carried out in two stages. First, sociodemographic and clinical data were collected during a household interview at the participant’s home. Then, weight and height were measured, urinary samples were collected, and a spirometry test was conducted for evaluation of pulmonary function during a subsequent visit to a mobile examination center. Full-time members of the Canadian Forces and Residents of Indian Reserves were excluded [22]. The Health Canada’s Research Ethics Board provided ethical approval for CHMS [23]. More details about recruitment strategies and assessment tools have been previously published [22,24].

The current analyses included participants aged 35 to 79 years old with complete physical activity and spirometry data. This age range has been chosen because multimorbidity prevalence drastically increases after 35 years [25]. Pregnant women and participants with functional limitations were excluded. [22,24].

Sociodemographic characteristics collected during interviews included age, sex, level of education, household income, working status, and marital status. Body mass index was calculated from weight and height measures. Active and passive smoking was characterized by urinary cotinine level. Smokers also reported age of first cigarette smoked, daily cigarette consumption, and number of daily smoking years [26]. The following clinical characteristics were self-reported with yes/no questions during interview: self reported diagnosis of COPD and current mood disorders, cough, phlegm, and shortness of breath. Participants reported their sleep duration. They also assessed their sleep problems, restorative sleep, and difficulty to stay awake with a Likert scale. Self-reported health, and mental health were self-reported with one item, using a 4-point response scale ranging from *poor* to *excellent*.

Diagnosis of three CDs (heart diseases [heart attack or disease], diabetes [type 1 or 2], and cancer [cancer or cancer survivor]) was self-reported. The questionnaire specified, “Remember, we are interested in conditions diagnosed by a health professional” [22]. A high association has been found between this self-report measure and physician diagnosis for these three CDs [27]. Pre-bronchodilator spirometry data were used to identify adults with COPD, which is frequently underdiagnosed in the general population [28]. Participants with a ratio of forced expiratory volume in one second (FEV1)/forced vital capacity (FVC) < 0.70 were classified as adults with COPD [29] (more details in Evans et al. [30]). For the current analyses, participants were grouped by number of CDs into 0, 1, 2, and 3 or more CDs.

The CHMS participants wore an Actical omnidirectional accelerometer (Phillips-Respironics) during their waking hours for 7 days. The accelerometer data collection started at the first occurrence of midnight after the mobile examination center appointment. Participants returned their monitor with a prepaid envelope. Data are not visible to participants while they are wearing the device. The omnidirectional accelerometer measures and records time-stamped acceleration in all directions, thereby providing the intensity of physical activity. The Actical data were summed with an interval of 1 min [31] and also translated into daily steps accumulated per minute. Esliger et al. [31] demonstrated that Actical accelerometer provided a valid measure of physical activity in adults [31]. According to Demeyer et al. [32] recommendations, a valid day of accelerometry was defined as >8 h of wear time. Participants with >4 valid days were included in analyses. The number of minutes per day spent in physical activity of different intensity levels was categorized using standard values of counts per minute (cpm) for adults: sedentary behaviors (<100 cpm), light physical activity (LPA) (100 to 1534 cpm), and MVPA (≥1535 cpm) [24]. We excluded study participants with extreme counts (i.e., 20 000 cpm) [33]. 

Four weighted analyses of covariance regression models (ANCOVAs) incorporating age, sex, body mass index, accelerometer wearing time (number of hours per day) and season [34], working status, self-perceived health, smoking (level of cotinine), self-perceived mental health, education level, and income was carried out [35,36]. Absence of CD was use as the reference category in our set of multivariate analyses. The following dependent variables were separately used in ANCOVAs: average daily minutes of LPA and MVPA, average daily steps, and average daily minutes of sedentary behavior. Poisson models were carried out for MVPA data because this outcome was not normally distributed. Analyses were carried out with R 3.3 and *survey* package [37]. We used weights (i.e., activity monitor subsample weights combining cycles 1, 2, and 3) and bootstraps provided by the CHMS for all statistical analyses.

Sensitivity analyses were carried out to examine the possible effects of each individual CD. Thus, we compared adults with a self-reported diagnostic of heart disease, cancer or diabetes with a control group for each physical activity outcomes and sedentary behavior. Results for adults with COPD have been previously published [38].

## 3. Results

### Participants

In total, we included 6270 CHMS participants in analyses (rounded to the nearest 10 as per Statistics Canada confidentiality requirements). Cross-sectional weighted analyses indicated that 23.9%, 4.9%, and 0.5% had one, two, and three or more CDs, respectively. Participants with multimorbidity averaged 9.7 (SE = 0.9) (2 CDs) and 6.4 (SE = 1) (≥3 CDs) minutes per day in MVPA and 171.4 (SE = 7) (2 CDs) and 155.4 (SE = 9.6) (≥3 CDs) minutes per day in LPA, and accumulated an average of 5779 (SE = 303.5) (2 CDs) and 5483 (SE = 666.9) (≥3 CDs) steps per day. Multimorbid participants spent an average of 579.7 (SE = 6.4) (2 CDs) and 606.8 (SE = 16.1) (≥3 CDs) minutes per day in sedentary behaviors. Detailed participant characteristics are shown in Table 1 and in the Appendix A. Descriptive data about physical activity outcomes and sedentary behavior among participants with heart disease, diabetes, cancer or COPD are also provided in the Appendix A [38].

Multimorbid participants had significantly lower daily MVPA (2 CDs [*p* = 0.02 × 10^−4^, *d* = −0.12]; ≥3 CDs [*p* = 0.07 × 10^−5^, *d* = −0.28]), LPA duration (2 CDs [*p* = 0.05 × 10^−1^, *d* = −0.07]; ≥3 CDs [*p* = 0.04 × 10^−3^, *d* = −0.24]) and daily number of steps (2 CDs [*p* = 0.02 × 10^−2^, *d* = −0.10]; ≥3 CDs [*p* = 0.01 × 10^−1^, *d* = −0.17]) compared to adults without CDs. Daily time spent in sedentary behaviors was significantly higher in multimorbid participants (2 CDs [*p* = 0.08 × 10^−1^, *d* = −0.07]; ≥3 CDs [*p* = 0.02 × 10^−1^, *d* = −0.17]) (Figure 1). No significant differences in MVPA, LPA, or number of steps per day were found between adults with a single CD compared to those with no CDs. Details about statistical findings are available in the Appendix A.

Sensitivity analyses showed that adults with heart disease had significantly lower levels of physical activity (for all outcomes) and spent more time in sedentary behaviors. Adults with cancer or diabetes had also a significant higher time spent in sedentary behaviors. No significant differences between the control group and adults with COPD was found. 

## 4. Discussion

This study suggests that multimorbid Canadian adults spend significantly less time per day in physical activity and more time in sedentary behaviors than adults without CDs. These findings are consistent with previous studies analyzing self-reported physical activity and sedentary behaviors measures [10,11,12,13,39]. It is important to note that differences were associated with small effect sizes. This can be explained in part by the inclusion of adults with risk conditions (e.g., hypertension, arthritis, insomnia) in the group with no CDs, which could impact the daily mean time spent in physical activity and sedentary behaviors for this group. These smaller effect sizes could be explained in part by a potential bias selection in previous investigations. Among studies using a convenience sample [40,41], investigators compared adults with CD with a control group of generally active and healthy adults (e.g., senior group practicing aerobic fitness). Consequently, the adjusted mean differences observed were higher.

No significant differences were found between adults with a single CD versus those with no CDs. This differs from previous research. Studies comparing physical activity levels in adults with heart diseases [42], diabetes [43], cancer [44], and COPD [40] concluded that adults with these CDs were significantly less active than controls without these diseases. These contradictory results may partly be explained by the use of self-reported physical activity measures (i.e., physical activity level is generally overestimated with questionnaires [45]).

Findings from sensitivity analyses suggested that the examined CDs were differentially associated with physical activity and sedentary behaviors. Heart disease was systematically related with lower physical activity outcomes, but more inconsistent findings were found among adults with other CDs. Thus, the lower level of physical activity identified in multi-morbid participants may be driven by the self-reported diagnostic of heart disease.

The weighted prevalence of multimorbidity in the current study was lower than in previous studies examining health outcomes in national samples. Systematic reviews show wide variability in multimorbidity rates, ranging from 13% to 72% in the general population [46]. This variation reflects differences among studies in the definition of multimorbidity used, the criteria used to define chronic diseases, the number of diagnoses assessed, and the sub-population considered [47].

Major strengths of this investigation include the analysis of objectively measured physical activity and sedentary behavior data, the use of a nationally representative sample of adults from a wide age range, and the classification of multimorbidity based on the four most prevalent CDs in Canadians. Absence of self-reported data about physical activity domains (i.e., leisure, active travel, work and household activities) in CHMS (2009–2013) [15] represents one limitation. These data could provide a clearer understanding of lower levels of physical activity in multi-morbid adults [48]. Another important limitation of the present study involves the reliance on self-reported diagnosis for three of the four characterized CDs. Moreover, the cross-sectional study design prevents any causal inferences between physical activity and sedentary behaviors with the progression of CDs. Finally, the CHMS did not include Canadians living in native reserves. This may modify our findings because native Canadian adults report a high level of physical inactivity [49].

Taken together, these findings underscore that the “move more and sit less” public health strategy [9], arguing in favor of interventions targeting physical activity and sedentary behaviors, should be developed or adapted for Canadian multimorbid adults. Future prospective research is required to assess the impact of physical activity and sedentary behaviors on the development of multimorbidity.

## Figures and Tables

**Figure 1 sports-07-00113-f001:**
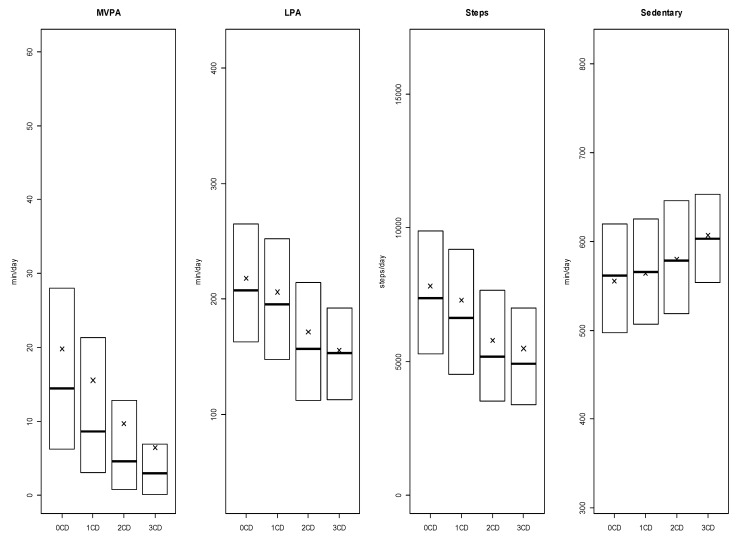
Weighted boxplots for daily minutes of moderate-to-vigorous physical activity (MVPA) and light physical activity (LPA), steps per day, and daily minutes of sedentary behaviors for adults with 0, 1, 2, or >3 Chronic Diseases (CDs). Note: Medians are represented by the bold lines, means are represented by the x-marks, 1st quarter and 3rd quarter are respectively bottom and top of boxes. Whiskers are excluded because Statistics Canada does not allow figures with individual data representation. CD = Accumulation of heart disease, diabetes, cancer and/or COPD.

**Table 1 sports-07-00113-t001:** Weighted characteristics of participants from cycles 1, 2, and 3 of the Canadian Health Measures Survey (CHMS) with accelerometry data.

	0 chronic Disease	1 chronic Disease	2 chronic Diseases	3 or More Chronic Disease
Age M(SE)	50.7 (0.3)	57.1 (0.5)	63.67 (0.7)	68.1 (1.3)
Sex (women) %(N)	52.1 (6,498,043)	49.8 (2,106,520)	45.4 (396,980)	28.2 (23,461)
BMI M(SE)	26.7 (0.17)	27.2 (0.3)	27.4 (0.5)	27.6 (0.9)
Marital status (alone) %(N)	22.8 (2,839,175)	30.4 (1,288,662)	33.0 (289,111)	26.6 (22,105)
Worked at job last year %(N)				
Yes	78.6 (9,843,658)	59.7 (2,508,893)	33.4 (317,453)	
No	19.7 (2,463,938)	34.4 (1,445,542)	56.2 (534,773)	
Study/retired	1.7 (210,661)	6.0 (250,236)	10.4 (98,708)	
Income %(N)				
<$15k	3.1 (389,603)	4.4 (184,754)	7.7 (67,706)	5.0 (4191)
$15k–$19.99k	1.9 (240,147)	3.1 (132,711)	6.5 (56,496)	6.0 (5005)
$20k–$29.99k	7.0 (879,349)	10.5 (444,517)	11.1 (96,851)	23.8 (19,777)
$30k–$39.99k	8.5 (1,058,326)	13.9 (588,064)	12.6 (110,048)	15.0 (12,509)
$40k–$49.99k	9.3 (1,156,616)	11.6 (490,457)	13.2 (115,780)	15.6 (12,930)
$50k–$59.99k	8.5 (1,057,049)	7.2 (303,850)	16.9 (148,125)	2.6 (2119)
$60k–$79.99k	15.4 (1,927,038)	15.5 (653,634)	10.4 (90,628)	12.6 (10,437)
$80k–$99.99k	12.5 (1,559,372)	10.2 (433,063)	5.7 (49,815)	9.6 (7994)
≥$100k	33.8 (4,217,903)	23.6 (999,096)	16.0 (139,690)	9.9 (8214)
Self-reported COPD %(N)	0.2 (25,064)	2.3 (98,029)	10.3 (98,554)	
Coughs phlegm regularly %(N)	9.3 (1,156,208)	15.6 (659,775)	21.1 (184,623)	29.2 (24,273)
Simple chores make short of breath %(N)	7.8 (974,235)	14.3 (605,945)	26.5 (231,898)	23.4 (19,444)
Self-rated health %(N)				
Fair/poor	9.2 (1,143,509)	15.6 (660,276)	30.6 (267,604)	33.5 (27,835)
(Very)good/excellent	90.8 (11,341,895)	84.4 (3,569,870)	69.4 (607,533)	66.5 (55,339)
Self-reported mood disorder %(N)	10.6 (1,319,094)	12.4 (522,529)	11.5 (110,559)	
Other physical or mental condition %(N)	20.8 (2,600,578)	17.9 (756,620)	21.9 (192,042)	19.9 (16,530)
**Sleep outcomes**				
Sleep duration (hours) M(Se)	7.0 (0.03)	7.0 (0.07)	7.3 (0.14)	7.0 (0.28)
Frequency of sleep problems %(N)				
Never/rarely/sometimes	78.7 (9,816,393)	74.0 (3,138,652)	73.5 (706,371)	
Most of the/all the time	21.3 (2,652,053)	26.0 (1,105,320)	26.5 (255,072)	
Restorative sleep %(N)				
Never/rarely/sometimes	41.1 (5,126,526)	37.0 (1,571,640)	41.5 (399,061)	
Most of the/all the time	58.9 (7,341,540)	63.0 (2,672,641)	58.5 (562,452)	
Difficulty staying awake %(N)				
Never/rarely/sometimes	95.6 (11,926,974)	94.0 (3,983,628)	92.3 (886,114)	
Most of the/all the time	4.4 (546,150)	6.1 (256,529)	7.8 (74,466)	
**Smoking variables**				
Smoking %(N)	16.1 (2,014,372)	29.7 (1,246,607)	28.5 (273,110)	
Age smoked first whole cig. (years) M(SE)	16.0 (0.2)	15.6 (0.2)	15.7 (0.6)	16.1 (1.5)
Age smoking every day (years) M(SE)	19.0 (0.2)	18.9 (0.5)	18.4 (0.5)	19.5 (1.7)
Number of cig. smoked/day when at least one cig/month M(SE)	6.1 (0.3)	6.7 (0.3)	6.3 (0.4)	7.4 (2.1)
Number of years smoked daily (former daily smokers) M(SE)	19.0 (0.2)	18.9 (0.5)	18.4 (0.5)	19.5 (1.7)
Levels of cotinine M(SE)	214.1 (22.6)	416.9 (36.8)	394.9 (102)	358.22 (138.2)
**Spirometry**				
FEV_1_/FVC M(SE)	0.78 (0.001)	0.71 (0.003)	0.70 (0.01)	0.65 (0.02)
**Characteristics of physical activity and sedentary behavior**				
Acc. wearing time (hours/day) M(SE)	13.8 (0.04)	13.6 (0.07)	13.1 (0.1)	13.2 (0.3)
MVPA (min/day) M(SE)	19.8 (0.7)	15.6 (1)	9.7 (0.9)	6.4 (1)
LPA (min/day) M(SE)	218.1 (2.6)	205.9 (4.2)	171.4 (7.1)	155.4 (9.6)
Steps per day M(SE)	7817 (104.6)	7291 (177.8)	5779 (303.5)	5483 (666.9)
Sedentarity (min/day) M(SE)	555.3 (2.5)	564.2 (4.2)	579.7 (6.4)	606.8 (16.1)

Note: In some cases, data for participants with 2 CDs and 3 or more CDs were merged for presentation of descriptive statistics to respect confidentiality requirements of Statistics Canada; Chronic disease = Accumulation of heart disease, diabetes, cancer and/or COPD; BMI = Body Mass Index; FEV = Forced expiratory volume; FVC = Forced vital capacity; MVPA = Moderate-to-vigorous physical activity; LPA = low physical activity.

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
