# Peer review of "Device-Assessed Physical Activity and Sedentary Behaviors in Canadians with Chronic Disease(s): Findings from the Canadian Health Measures Survey"

_sports, 2019, doi:10.3390/sports7050113_

Round 1

Reviewer 1 Report

This study aimed to investigate differences in MVPA, LPA, steps per day, and SB between adults with and without chronic diseases in a large representative sample of Canadians by use of accelerometry. I find the study well conducted. Please find some specific areas of improvement below.

Previous studies using both subjective and objective measurement of physical activity in this field is introduced in the introduction. As the lack of objective measures of physical activity is an important rationale, do previous results differ between studies using differing methodology? Due to less error, it would be expected that accelerometry would be superior to subjective methods. If this is the case, it would strengthen the rationale that we need more studies using objective measurement.

Line 43-44. It is unclear that the definition is given for SB. Please clarify that this definition is for SB, which differs from inactivity.

The description of the sample can be improved, although a previous publication that describes recruitment and sampling is cited.

Line 114-120. Accelerometer cut points is reported twice. Please remove.

How is chronic diseases analyzed? Was dummy coding used? Specify that 0 is used as the reference category(?). This variable is not stated as one of the independent variables line121-124. Please include. I recommend the analyses could be extended to also analyze each single condition, possibly to support and nuance why results contrast earlier studies (discussion line 165). Could also inclusion of other diseases, like mental or musculoskeletal conditions, make a difference? Could the contrasting results to previous studies be a result of over adjustment, for example by work status, which is strongly related to the number of diseases?

It is not perfectly clear for me which independent variables were included in the statistical models. Is self-rated health and mental health single variables, or many variables, as reported in table 1? Consider whether all variables reported in table 1 is necessary.

Why is results not reported for work status of those having 3 or more conditions? If it is to protect confidentiality, why is it reported differently from other variables where this is the case?

Please specify how effect sizes are calculated.

Figure 1: I suggest revising the scale of the figure. Please also explain what the bold lines, the x-marks, and boxes indicate.

Reporting of p-values with more than 3 decimals is redundant. Please revise to improve readability.

Line 169. I agree that questionnaire generally may overestimate PA levels, but is PA levels differentially overestimated by the groups? A similar overestimation would result in a similar differences between groups. Moreover, how could bias influence the findings (in cross-sectional studies)? The discussion on why results differ from other studies should be extended to provide new insight.

The use of accelerometers is a strength to the study, but they also have limitations that should be acknowledged in the discussion line 179-.

Author Response

Thank you very much for reviewing, here are the answers. Note that little extra changes were made in the manuscript to respect a duplicate check.

Reviewer 2 Report

Using the 2007-2013 Canadian Health Measures Surveys, the authors aimed to estimate accelerometer-measured physical activity and sedentary behavior by the number of comorbidities individuals had at the time of the survey. The authors found that Canadians with more comorbidities engage in less physical activity and more sedentary behavior. They use a large, representative sample and objective measures of physical activity and sedentary behavior to execute their study.

Introduction:

Although detailed, the authors fail to argue the novelty and what their findings will add to the existing literature in the introduction.

Methods:

Not all variables discussed in the methods section were used in the models. How were confounders chosen to include in the model? For example, why was sleep not considered a confounder? 

To strengthen the justification for using self-reported measures of disease, authors should cite reliability or validity studies of the self-report of the comorbid conditions you chose. 

There is no definition of heart disease. Did this include stroke, MI, heart failure? If so, an individual may have all three and have more than one comorbidity. 

The authors chose not to include hypertension, arthritis, etc. as comorbid conditions but did not provide justification as to why. They do not even take into account these conditions in the model. 

Results:

How many individuals were excluded from the study? The authors present a final sample size, but do not explain how many were excluded for not participating in accelerometry data collection. This number has implications for selection bias (which is also not mentioned in the limitations). 

The main results were not discussed in the results section and thrown into a supplementary file. Why were these not included in the main results of the paper? 

Discussion:

Authors did not discuss the limitations of accelerometer-measured physical activity and sedentary behavior. Accelerometers do not take into account context of the activity or sedentary behaviors and may limit interpretation of the findings.

The limitations section should be elaborated and discussed further. The authors do not include all potential CDs and should elaborate on why and how they tried to mitigate that. Limitations of the overall study design should also be included such as excluding Indian Reserves, Crown lands, and remote regions. 

Tables and figures:

All tables need footnotes with acronyms and definitions, particularly what diseases were included in the definition of you CDs. Tables and figures should stand alone. 

Figure one requires a legend. It is unclear what figure 1 is showing without context. 

Author Response

(The authors gave the same response as above.)

Reviewer 3 Report

Thank you for taking the time to prepare and submit this manuscript for review and thereby providing me with the opportunity to read your work.  I have made a few comments below for your consideration

Lines 43-44 - Definition has been updated slightly since 2012.  Please see 2017 publication here: https://ijbnpa.biomedcentral.com/articles/10.1186/s12966-017-0525-8.  Citation: Tremblay MS, Aubert S, Barnes JD, Saunders TJ, Carson V, Latimer-Cheung AE, Chastin SFM, Altenburg TM, Chinapaw MJM, SBRN Terminology Consensus Project Participants. Sedentary Behavior Research Network (SBRN) – Terminology Consensus Project process and outcome. Int J Behav Nutr Phys Act. 2017 June 10;14(1):75.

Line 81 – can a statement be added to clarify why this age range in under investigation? Please can this go beyond that this in the age range of the Canadian Health Measures Survey.

Lines 117-120 – duplication of sentence already provided in this paragraph.  Please delete.

Results - Why were MVPA presented in combination alone?  There is an argument (and research evidence to support) that VPA would accrue greater health benefits, therefore, it is recommended that analyses be repeated and reported for MPA and VPA separately.  This could bring an added dimension to the results and discussion sections.

Discussion - Is there further criticism of the CHMS?  Certain riskful behaviours associated with NCDs are not included in this paper (e.g., alcohol consumption, eating behaviours), and therefore could this be a recommendation for future research?

Author Response

(The authors gave the same response as above.)

Round 2

Reviewer 1 Report

I thank the Authors for providing appropriate answers to most of my comments and making relevant changes to their manuscript. I only have one more point to consider.

Relating to my previous comment on emphasizing the benefit of including objective measures, what is more important than bias in this case, is random error that attenuates associations. I suggest including the study by Lee et al. "Accelerometer-Measured Physical Activity and Sedentary Behavior in Relation to All-Cause Mortality the Women's Health Study." Circulation 137, no. 2 (Jan 2018): 203-05, which strongly indicates that better measures of physical activity strengthen associations with health outcomes.

Author Response

Thank you very much for the reviewing. Here are the changes.

Reviewer 2 Report

I originally asked the authors to strengthen the justification for using self-reported measures of disease, authors should cite reliability or validity studies of the self-report of the comorbid conditions you chose. 

AUTHORS RESPONSE: # For the heart disease, diabetes and cancer, the self-reported measurement was assessed with the following sentence : “Remember, we are interested in conditions diagnosed by a health professional”

We added this sentence in the method section. For the COPD, we used spirometry data to determine the COPD status.

Although the authors are interested in conditions diagnosed by a health professional, they are still relying on self-report of these conditions and MUST cite validity or reliability studies.

Author Response

(The authors gave the same response as above.)
